# communications
# earth & environment

# Global survey shows planners use widely varying sea-level rise projections for coastal adaptation

Daniella Hirschfeld [1✉], David Behar[2], Robert J. Nicholls [3], Niamh Cahill [4,5], Thomas James [6], Benjamin P. Horton [7,8], Michelle E. Portman[9], Rob Bell [10,11], Matthew Campo [12], Miguel Esteban [13], Bronwyn Goble[14], Munsur Rahman[15], Kwasi Appeaning Addo [16], Faiz Ahmed Chundeli [17], Monique Aunger[18], Orly Babitsky[9], Anders Beal[19], Ray Boyle[20], Jiayi Fang [21], Amir Gohar[22], Susan Hanson [3,23], Saul Karamesines[1], M. J. Kim [24], Hilary Lohmann[25], Kathy McInnes [26], Nobuo Mimura [27], Doug Ramsay[28], Landis Wenger[1] & Hiromune Yokoki[29]

Including sea-level rise (SLR) projections in planning and implementing coastal adaptation is crucial. Here we analyze the first global survey on the use of SLR projections for 2050 and 2100. Two-hundred and fifty-three coastal practitioners engaged in adaptation/planning from 49 countries provided complete answers to the survey which was distributed in nine languages – Arabic, Chinese, English, French, Hebrew, Japanese, Korean, Portuguese and Spanish. While recognition of the threat of SLR is almost universal, only 72% of respondents currently utilize SLR projections. Generally, developing countries have lower levels of utilization. There is no global standard in the use of SLR projections: for locations using a standard data structure, 53% are planning using a single projection, while the remainder are using multiple projections, with 13% considering a low-probability high-end scenario. Countries with histories of adaptation and consistent national support show greater assimilation of SLR projections into adaptation decisions. This research provides new insights about current planning practices and can inform important ongoing efforts on the application of the science that is essential to the promotion of effective adaptation.

[1] Department of Landscape Architecture and Environmental Planning, Utah State University, 4005 Old Main Hill, Logan, UT 84322-4005, USA. [2] San Francisco Public Utilities Commission, San Francisco, CA, USA. [3] Tyndall Centre for Climate Change Research, University of East Anglia, Norwich, UK. [4] Department of Mathematics and Statistics, National University of Ireland, Maynooth, Ireland. [5] Irish Climate Analysis and Research UnitS (ICARUS), Maynooth University, Kildare, Ireland. [6] Geological Survey of Canada, Natural Resources Canada, Victoria, Canada. [7] Earth Observatory of Singapore, Nanyang Technological University, Singapore, Singapore. [8] Asian School of the Environment, Nanyang Technological University, Singapore, Singapore. [9] MarCoast Ecosystems Integration Lab, Technion – Israel Institute of Technology, Haifa 32000, Israel. [10] Bell Adapt Ltd, Hamilton 3210, New Zealand. [11] Environmental Planning Programme, School of Social Sciences, University of Waikato, Te Whare Wananga o Waikato, Hamilton, New Zealand. [12] Edward J. Bloustein School of Planning & Public Policy, Rutgers, The State University of New Jersey, New Brunswick, NJ, USA. [13] Department of Civil and Environmental Engineering, Waseda University, Tokyo, Japan. [14] The Oceanographic Research Institute, Durban, South Africa. [15] Institute of Water and Flood Management (IWFM), Bangladesh University of Engineering and Technology (BUET), Dhaka 1000, Bangladesh. [16] University of Ghana, Accra, Ghana. [17] School of Planning and Architecture, Vijayawada, Andhra Pradesh, India. [18] Geological Survey of Canada, Lands and Minerals Sector, Natural Resources Canada 601 Booth Street, Ottawa, ON, Canada. [19] Woodrow Wilson International Center for Scholars, Washington, DC, USA. [20] College of Environmental Design, University of California Berkeley, Berkeley California, USA. [21] Institute of Remote Sensing and Earth Sciences, School of Information Science and Technology, Hangzhou Normal University, Hangzhou 311121, China. [22] University of the West of England, Bristol, UK. [23] Faculty of Engineering and Physical Sciences, University of Southampton, Boldrewood Campus, Burgess Road, Southampton, UK. [24] Ministry of Oceans and Fisheries affairs, Busan, Republic of Korea. [25] Department of Planning and Natural Resources, St. Croix, USVI, USA. [26] Climate Science Centre, CSIRO Environment, Aspendale, VIC, Australia. [27] Global and Local Environment Co-creation Institute, Ibaraki University, Ibaraki, Japan. [28] National Institute for Water and Atmospheric Research, Auckland, New Zealand. [29] Department of Civil, Architectural, and Environmental Engineering, Ibaraki University, Ibaraki, Japan. ✉email: Daniella.hirschfeld@usu.edu

The appropriate use of sea-level rise (SLR) projections in coastal decision-making is critical but challenging. The scenarios used and their application will have profound impacts on our social, ecological, and economic coastal systems[1–3]. Hundreds of millions of people currently living in coastal zones face significant risks due to SLR, and the implementation of proactive adaptation measures would be prudent[4]. Coastal ecosystems are already under stress from ocean warming, acidification, and SLR, compounded by human interventions, and expected responses over this century include habitat contractions, translocation, and loss of biodiversity and functionality[5]. Recent estimates suggest that coastal adaptation costs for the developing world will range from $26–89 billion a year by 2040 s[6]. Hence, the SLR scenarios used by decision-makers have substantial cost and risk implications, with the danger of overinvestment for unnecessary protection[7] or underinvestment, leading to escalating inundation risk and emergency response challenges for vulnerable communities[8,9].

Sea-level science is a well-developed field of study with decades of scientific experience with increasing sophistication and new modeling platforms providing a deeper understanding of future sea levels and associated uncertainties[2,10]. The Intergovernmental Panel on Climate Change (IPCC) has released six major assessments, based on an extensive body of literature[2,11]. Researchers have broadened work from a focus on median SLR estimates to the consideration of high-end SLR scenarios, including increasing frequency of flooding, changing storm events, and waves, to capture the widening uncertainty[2,11–15]. Global emissions in the coming decades and the sensitivity and tipping points of various SLR components drive uncertainty in projections, especially for the Greenland and Antarctic ice sheets[16–18]. This widening uncertainty challenges decision analysis[14,19]. Assimilation by practitioners, managers, and decision-makers of long-term SLR requires recognition and a clear understanding of the range of uncertainties and how they can be articulated in planning[20–22].

Coastal and estuarine environments are highly dynamic, and communities living within them have a long history of adaptation[23,24]. Formal efforts to build a shared body of knowledge including frameworks to address SLR adaptation began with the first IPCC assessment and associated guidance in the 1990s[25–30]. Regional and local efforts to plan for future climatic conditions and implement adaptation measures have been undertaken by coastal managers for the last two decades and these efforts are still growing[31–33]. Increasing knowledge[34], public awareness, and programs to facilitate and promote adaptation[35] in some places puts pressure on decision-makers to incorporate sea-level science into planning efforts and guidance[23,36,37].

Successful coastal adaptation requires robust science-policy integration and well-designed climate services, both built on ensuring the usability of scientific information[38,39]. Building and designing these systems requires an understanding of how to make science-based decisions in the context of increasing uncertainties in SLR over time. With a few exceptions[40,41], there has been little assessment of adaptation practice in coastal areas, especially of the sea-level scenarios used by practitioners to inform the science-policy interface. Assessment of sea-level adaptation practices and accompanying scenarios will inform the future development of sea-level science and would be accompanied by an improved understanding of how to translate uncertainty in sea-level projections into the decision environment.

Here we distributed the first global survey on this topic via a confidential questionnaire to coastal practitioners in every inhabited continent; we provided the questionnaire in English and translated it into eight additional languages. The questionnaire asked for specific time horizons and projection information currently used in coastal planning materials for areas under their jurisdiction, the science behind SLR projections used in policy, and how practitioners apply SLR projections. Through quantitative and qualitative analyses, we found spatial relationships between global coastal regions and the degree of use of sea-level science in plans. We found surprisingly that most coastal managers are using a single SLR projection rather than considering a range of possible SLR values to account for uncertainty. We also learned that a wide range of future projections are in use revealing that there is no globally standardized approach to selecting and using SLR projections.

## Results

**Uneven distribution of the application of sea-level science.** We gained important insights at the global, continental, regional, and country scales about whether and to what degree coastal managers are using SLR projections in their coastal planning. Working closely with partners and using a snowball sampling approach[42] 253 coastal managers completed our questionnaire. This sample represents the first global data collection on SLR use in decision making[43] (Supplementary Table 1, Supplementary Table 2 and Supplementary Fig. 1). Our respondents all identified as planners working primarily (89%) for local governments (e.g., cities, councils, municipalities, towns, and native settlements) and sub-national governments (e.g., districts, provinces, regions, states, and territories). Our analysis focuses on the information provided by our respondents about the use of sea-level science, not on the number of respondents per region. The distribution of responses in our samples, however, is clearly geographically uneven, which contributes to the fact that we did not note a strong correlation between the use of future sea levels in planning and country-level covariates including GDP, education levels, and the human development index. That is not to say that no relationship exists but rather that further research with a different sampling approach and a greater number of respondents could better explore such relationships.

We found that 181 (72%) respondents are in Group 1, defined as having formally adopted guidance materials, reports, or policy documents that include SLR projections in their coastal planning processes. This group represents areas with nearly half of the world's coastal population. We also found that 67 (26%) respondents are in Group 2 and are trying to use SLR projections; however, they do not have a formal policy in place yet. Finally, only 5 respondents (2%) are in Group 3 defined as not currently working with SLR projections in their planning (Supplementary Table 3), possibly reflecting in part the non-response of planners who are not considering SLR to a questionnaire focusing on SLR.

At the continental scale (Fig. 1A and Supplementary Table 4), we found that Europe, Australia/Oceania, and North America were the continents with the largest proportions of respondents using SLR projections in planning. Respectively, they had 87% ($N = 31$), 84% ($N = 44$) and 77% ($N = 126$) of their respondents in Group 1. The continents with the lowest percentages of respondents in Group 1 are Asia and South America (36% ($N = 39$) and 33% ($N = 3$), respectively). Africa is intermediate, with 50% ($N = 10$) of respondents in Group 1. Regionally (Fig. 1B and Supplementary Table 4), we observe important differences within continents. In Europe we found that North and West Europe have 95% ($N = 20$) of their respondents in Group 1, compared to only 50% ($N = 6$) in Southern Europe (Northern Mediterranean). Continentally aggregated data obscures the North America dichotomy between the United States, where 80% ($N = 95$) of respondents are in Group 1, and the Caribbean Islands, where only 20% ($N = 5$) are in Group 1.

We found that certain countries are particularly high users of SLR projections in their coastal planning processes

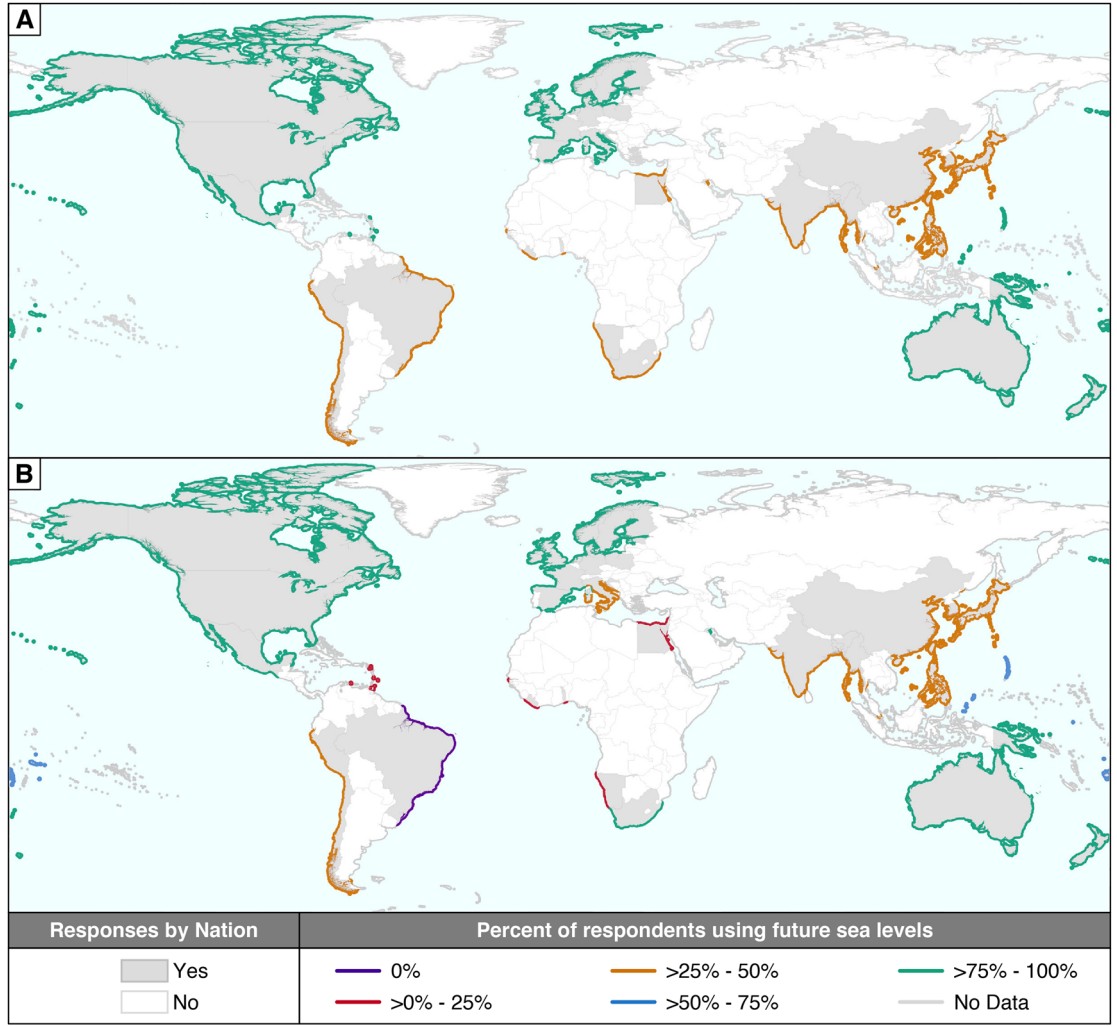

| Responses by Nation | | Percent of respondents using future sea levels | | | |
|---|---|---|---|---|---|
| | Yes (gray) | —— 0% | —— >25% - 50% | —— >75% - 100% | |
| | No (white) | —— >0% - 25% | —— >50% - 75% | No Data | |

**Fig. 1 Study scope and use of sea-level rise projections in planning.** Percent of respondents by continent (**A**) and coastal region (**B**) who are using sea-level rise in coastal planning processes and the countries (in gray) that provided responses. See Supplementary Table 2 for a list and details of coastal regions.

(Supplementary Table 4), such as New Zealand (90% of respondents, $N = 10$). This reflects the availability of SLR scenarios in clearly articulated guidance for practitioner use created at the national level[37]. In another example, we found that in the United Kingdom, which has a long history of including relative SLR in infrastructure design pre-dating climate change concerns (e.g., Gilbert & Horner[44], 1984), 100% of respondents ($N = 8$) use SLR projections in their planning processes. We infer from these examples that robust national guidance and a longer history of SLR integration in planning contribute to the ongoing use of SLR projections in current coastal planning.

In contrast, we found certain regions and countries to have a low use of SLR projections (Supplementary Table 4). Japan, where 80% of respondents ($N = 5$) reported not using SLR projections in planning has an extreme tsunami risk as demonstrated in 2011[45]. This extreme risk and recent experience, including rebuilding and adapting to tsunami risk, may overpower concerns about smaller SLR projections of between 1 and 2 meters. However, tsunami risk greatly increases with SLR and therefore SLR ought not to be ignored[46]. Note that coastal management policies change over time. Japan's coastal management policy has changed since the survey for this study was performed. The Ministry of Land, Infrastructure, Transport, and Tourism revised the Basic Policy for Coastal Conservation under the Coastal Act in November 2020 to incorporate SLR. The new Basic Policy gives

a firm guidance to local governments when they revise the basic plan for coastal protection and land use. Other places, such as Western Africa, where none of our respondents said that SLR is part of planning, could be hindered by a lack of capacity for long-term planning (e.g., 2100 and beyond) and rather focus on the near term (i.e., next 10–20 years). These findings suggest that lack of capacity and competing priorities could both be playing a role in areas with limited use of future SLR projections.

**The data structures used by planners to depict sea-level rise futures.** We asked coastal managers if SLR projections fell under four formal data structures (A, B, C, D) for both 2050 and 2100. Of the 143 respondents (56% of the original sample) that indicated use of these formal structures, the most common structure (A) is a singular estimate, which is used by 76 (53.1%) respondents (Fig. 2). A low, intermediate, and high estimate was the second most common structure (C) used by 28 (19.6%) respondents, while 20 (14.0%) respondents used a low and high estimate (B). The least common structure (D), with 19 (13.3%) respondents, was the structure with a low, intermediate, high, and high-end estimate. The latter was defined as the highest future sea-level estimate based on extreme but plausible information, which in some jurisdictions is referred to as H++[47]. In addition to these four common structures, forty respondents (16.0% of the original sample), are using unique structures tailored to their locations.

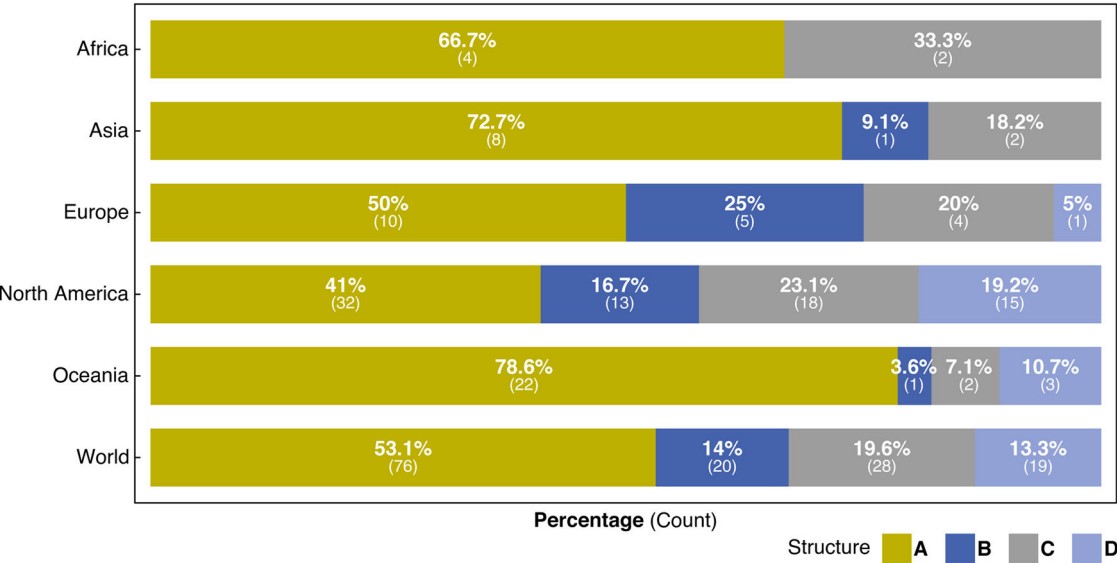

**Fig. 2 Structures of sea-level rise projections used globally.** Respondents formally structure the use of sea-level rise projections for planning purposes in four ways: **A** is a singular estimate, **B** is a low and a high estimate, **C** is a low, intermediate, and high estimate, and **D** is a low, intermediate, high, and high-end estimate. Shown are aggregated responses for five distinct geographical regions and the globe.

Notably, of the respondents that rely on these formal structures, Structure A is used by the majority on every continent. Not all 143 respondents that use the formal structures gave a projection for 2100. The total in 2100 = 135 (which is what the numbers in Supplementary Table 5 sum to). In Oceania, Asia, and Africa Structure A is used by 78.6%, 72.7%, and 66.7%, respectively. This finding contrasts both with some guidance on planning for future SLR[14,19,37,48,49] and with the work of the scientific community to refine and clarify the range of future sea levels and associated uncertainties[2]. However, a single number is eventually needed in many contexts, especially by engineers designing coastal infrastructure. This number should arise out of careful consideration of a range of projections during the asset life cycle including high-end estimates for risk-averse decisions[50] and timing windows to exceed design thresholds[51]. We recognize that the use case[52] of our respondents would shed further light on the structures and selected projections. In our study, we find that respondents are applying the structures and the projections in many use cases (Supplementary Fig. 2). We also find that our respondents rely on many criteria to determine the right projections for their land-use (Supplementary Fig. 3) and infrastructure planning (Supplementary Fig. 4). Thus, we are not able to discern a relationship between the application type and the projections or structures used.

Some coastal managers in the United States, Northern/Western Europe, New Zealand/Australia, and Northern Africa are using a high-end SLR scenario (Structure D) (Supplementary Fig. 5). No other coastal regions in our sample are using this structure. The United States has the greatest use of high-end SLR scenarios, with 17 locations across the country using this type of scenario. The use of high-end SLR scenarios in plans provides an opportunity to understand the uncertainty, consider plausible high-end scenarios, and stress-test long-term adaptation options to better bracket and plan adaptation and avoid maladaptation[11]. Conversely, adoption of this extreme value in planning can lead the public and policy makers to mistakenly anticipate more expensive and socially disruptive adaptation measures than may be necessary[37,41,48]. To navigate these advantages and disadvantages to high-end SLR use, practitioners would benefit from more guidance concerning the use of high-end scenarios (see van de Wal, et al. 2022[50]).

We observe an interesting difference between Canada and the USA. In Canada 16 places (84%) are using a single future estimate (A) and 3 places (16%) are using low, intermediate, and high (C) SLR projections. Conversely, in the United States, there is a much wider range of approaches: 14 places (24%) are using a single future estimate (A), 11 places (19%) are using a low and high estimate (B), 16 places (28%) are using a low, intermediate, and high estimates (C), and 17 places (29%) are using a low, intermediate, high, and high-end estimate (D). This difference is likely the result of national and regional guidance that emphasizes or de-emphasizes high-end estimates. For example, the State of California explicitly calls attention to the H + + scenario of 3 meters in 2100 and recommends its use in extreme risk-averse decision contexts[53]. In contrast, British Columbia, where the majority of Canadian respondents work, recommends consideration of 1 meter of sea-level rise at 2100 and 2 meters at 2200, adjusted for vertical land motion[54].

**No global standard**. Our findings indicate that a wide range of future projections are used by coastal managers to plan for SLR in both 2050 (Supplementary Fig 6 and Supplementary Table 5) and 2100 (Fig. 3 and Supplementary Table 5). Here we focus on the sea-level rise projections for 2100 used by 135 respondents in the four scenario structures defined above. We report numbers rounded to the nearest centimeter. The Supplementary Tables provide more precise numbers. For Structure A (N = 71) the median is 0.90 m, with a minimum of 0 m in eight locations globally and a maximum of 2.03 m in Hayward, California in the United States. For Structure B (N = 19) the median low value is 0.61 m and the median high value is 1.40 m. For Structure C (N = 26) the median low value is 0.42 m, the median intermediate value is 0.71 m, and the median high value is 1.21 m. For the 19 respondents using Structure D the median low value is 0.53 m, the median intermediate value is 1.19 m, the median high value is 1.91 m, and the median high-end value is 3.05 m. We observe that the values for those using Structure A cover almost the full range of values from structures B and C, indicating that this approach is not limited to median or low-end estimates. Finally, we did not find a robust statistical difference between the structure used and median projections; however, those using Structure D do have a higher median for their high estimate

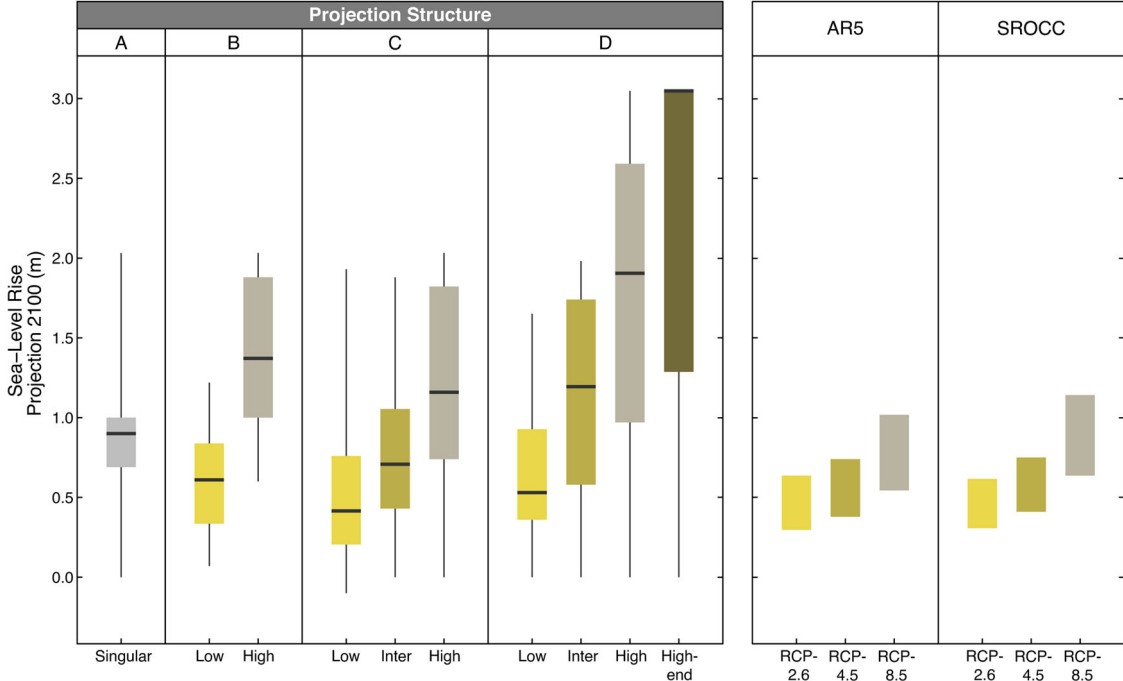

**Fig. 3 Comparison of sea-level rise projections in planning and available science.** Left: The SLR projections (in meters) for 2100, which respondents use in their coastal plans and guidance documents. Projections are grouped by the four projection structures (**A** to **D**) shown in Fig. 2 and shown as box plots with median values as the dark center line, the box representing the 25th to 75th percentiles, and the whiskers showing the full range of survey responses. Right: The IPCC Fifth Assessment Report (AR5)[1] and Special Report on the Ocean and Cryosphere in a Changing Climate (SROCC)[5] global projections show the "likely" ranges between the 17th and 83rd percentile.

and have adopted a median high-end estimate that exceeds projections used in Structures A, B, and C.

We compared the projections provided in the survey with IPCC Fifth Assessment Report (AR5)[1] and Special Report on the Ocean and Cryosphere in a Changing Climate (SROCC)[5], which are trusted global sources of SLR relevant to the timing of the survey (Fig. 3). Interestingly, many of the reported future sea levels for planning out to 2100 are lower or significantly higher than the range provided by AR5[1] and SROCC[5]. In total, we received 119 responses above the RCP8.5 scenario of 0.98 m across all scenario types (see Supplementary Table 7). This variation may reflect respondents following regional guidance that suggests higher (or lower) SLR than the global IPCC projections based on the timing of the guidance, known regional variations, the use of relative SLR, or the inclusion of larger amounts of projected sea-level rise that were given low confidence by the IPCC.

## Discussion

We present evidence from the first global survey of coastal managers on the use of SLR projections in planning that practitioners are incorporating SLR projections in decision making. However, we find evidence of a potential overreliance on singular estimates and highly inconsistent approaches to the selection of SLR projections. Singular estimates are appropriate and even necessary at the later stages of planning; however, it is currently best practice for SLR planning to include multiple scenarios, generally corresponding to different possible climate futures (climate scenarios), combined with advice on risk-based robust adaptation methods[14,19,37,48,49]. We also acknowledge that developing best-practice multiple SLR scenarios may pose a challenge for some jurisdictions with less adaptive capacity[55] and therefore recognize the important role played by climate service

providers including boundary organizations and government actors[56,57].

Our sample spans the globe with respondents from every habitable continent; however, we acknowledge that our responses do not align with global populations and are dominated by North American (49.8%) respondents. This imbalance could be an indication of location-specific factors inhibiting responses such as cultural and privacy differences in responding to questionnaires and lack of resources to respond[58]. Another contributor to response rates could be different vulnerabilities[4], with some places overwhelmed by existing threats unable to respond and other places not perceiving their future vulnerabilities and thus not motivated to respond. Future work is required to increase sample diversity, to better understand harder-to-reach parts of the globe, and to support adaptation in vulnerable communities possibly disadvantaged by capacity issues.

The decision context of the practitioners we surveyed could be a significant driver of the differences we observed[52]. For example, practitioners could be responsible for the construction of expensive long-lived infrastructure and therefore would more likely be risk averse. On the other hand, they could be responsible for designing a public park and could be less risk-averse. These two different risk scenarios would warrant different selections of SLR projections. Similarly, some respondents could be focused on short-term decisions, such as beach nourishment, while others could be responsible for long-term land-use decisions. These two groups would be using different SLR values. Respondents in the application section of our survey identified most use cases (Supplementary Fig. 2) and many different criteria for differentiating between projections (Supplementary Fig. 3 and Supplementary Fig. 4). Thus, we cannot match their use cases with the projections they provided. Survey design improvements would allow us to better link specific decisions with standardized structures and future SLR

projections. Future research should focus on understanding the decision contexts with a particular focus on risk and planning horizon concerns.

Another possible driver of the differences we observed is the source of the SLR projections used by different practitioners. For example, some practitioners could be relying on state guidance that is particularly risk-averse and uses a high-end value, and others could be relying on research that is focused on clarifying mean values. In our survey, we asked practitioners how the SLR projections in their plans were developed? Respondents identified three primary sources: (1) selected from projections developed by a higher level of government; (2) co-produced between scientists and practitioners; and (3) generated as guidance by an authority (Supplementary Fig. 7). We recognize that practitioners could be relying on regional or local projections rather than the global-mean projections and this could lead to differences. For example, the Atlantic coast of the United States is more vulnerable than other parts of the country due to subsidence from glacial isostatic adjustment[59]. Subsequent work is needed to more carefully examine the sources used by practitioners and the relationship between those sources and the original scientific research.

Our findings reflect the respondents' interpretation of the questions we posed. We asked for sea-level rise values used for planning purposes. Respondents could have understood these values to include additional water height contributors such as storm surge, regional sea differences, and vertical land motion or they could have understood the value to be the global value that was then adjusted to local conditions. Thus, for some cases, we may be comparing differing realizations of flood levels to the projections of mean sea-level change provided in the IPCC reports. However, here we consider what coastal managers understand to be the future projections for which they are designing and planning. Investigation of the documents provided by survey respondents could provide further insight. Future versions of this survey should structure questions in such a way as to get greater clarity from survey respondents. A future survey could request, in addition to sea-level guidance used by the respondent, plans that were developed based on the guidance, (e.g., The Bangladesh Delta Plan[60]). This would provide further information on guidance usage.

A further step in this line of research could be to assess whether and how certain larger-scale SLR guidance is assimilated into decisions. Specifically, does the design and regulatory environment of national guidance directly influence the local (i.e., city/county) level SLR planning? For example, does the national guidance in New Zealand, based on a dynamic adaptation pathway planning approach[37], provide local practitioners with more usable information? Additionally, more work is needed to understand the reasons behind the different approaches and progress in different communities. Interviews with practitioners across the globe would provide significant insights into the barriers encountered and opportunities available. More research is needed on how these policy and guidance documents inform the physical infrastructure and land-use planning decisions made by coastal managers.

As global sea levels continue to rise, planning, designing, and building resilient communities will become a more pressing societal challenge. Our research provides global data on how coastal practitioners use sea-level science in the adaptation planning of coastal lowlands. Consistent with past research on climate services, we find significant reliance on singular estimates (Fig. 2) and highly inconsistent approaches to assimilating sea-level science into decision-making (Fig. 3). This persistent disconnect raises concerns about coastal managers' ability to translate complex and uncertain futures into adaptation decisions. This is particularly true when coastal managers are using

high-end SLR scenarios, which are well suited to constrain adaptation options and understand the uncertainty but can be misapplied and lead to more expensive and socially disruptive adaptation measures than may be necessary. The literature indicates that high-quality translation services and peer learning through collaborative organizations improve practitioner use of sea-level science[20,61]. As the implementation of SLR adaptation strategies is becoming more prevalent, we hope that this assessment triggers similar and improved studies on the application of SLR science. The insights will create better bridges and shared understanding between science and coastal managers. Further improved surveys of the type described here are essential to inform and assess these efforts.

## Methods

**Recruitment and sample**. To understand the nature and extend of sea-level science assimilation into decisions on adaptation for coastal lowlands (e.g., land-use planning, infrastructure design, managed retreat), we recruited coastal managers from every habitable continent using a combination of two sampling methods. Two-hundred and fifty-three managers responded.

We used a snowball sampling approach to reach as many geographic locations as possible. This sampling technique is ideally suited to circumstances where it can be difficult to adequately define the sampling frame[62]. We asked collaborating researchers and climate change specialists at national and regional levels to provide names and contacts at more localized jurisdictions that were known to be involved in sea-level rise (SLR) planning. We also identified cities conducting SLR planning and then targeted relevant contacts directly within the city. To identify cities we used previous publications about SLR plans and websites (e.g., Climate Adaptation Knowledge Exchange, U.S Resilience Tool Kit, etc.) that provide case studies on SLR planning and design applications. For each location, one point of contact was identified from the official website or personnel database for that city. Some of the participants initially identified were not appropriate contacts due to organizational differences, retirement, or other factors. In these cases, the person usually provided replacement contacts.

To improve sample diversity, we used all five of the methodological recommendations articulated in Kirchherr & Charles[63]. First, the team relied heavily on personal contacts with each regional lead sending the same requesting email to their contact list requesting coastal managers at the local and regional scale. Second, we had a diverse seeding process reaching out to multiple contacts in a single region. For example, in the United States we reached out to both state-level officials at both the coastal zone manage agency and at the sea grant offices. Third, we worked hard to develop trust with individuals to get referrals for respondents. We made personal phone calls to certain contacts to help gain explain the research and enable their participation. Fourth, we were very persistent sending contacts multiple emails from both the online survey tool and the personal contact directly. Additionally, in some cases we worked with a team of contacts in a place to help ensure that the survey was completed. Fifth, we had two sampling waves and did a focused follow-up with people in regions that were hard to reach. Beyond these five methodologies, we also allowed for a range of ways to respond. Although we emphasized the online survey tool, we also allowed people to complete a PDF questionnaire at their office or over the phone with a researcher. Despite these efforts, we acknowledge that our sample has gaps and lacks the diversity we aspired to. Future research would benefit from greater reliance on alternative data collection methods to a survey instrument. Interviews would likely yield a greater sample diversity and more responses from regions where surveys and emails are unfamiliar and hesitation to participate is high.

Our respondents represent the first global data collection on SLR use in decision making. Though they were not evenly divided, no single continent represented over 50% of respondents (Supplementary Table 1). They comprised 10 (4.0%) from Africa, 39 (15.4%) from Asia, 31 (12.3%) from Europe, 126 (49.8%) from North America, 44 (17.4%) from Australia/Oceania, and 3 (1.2%) from South America. At the regional scale, North America Atlantic Ocean and North America Pacific Ocean had the greatest representation with 78 (30.7%) and 42 (16.6%) of the respondents, respectively. Pacific Ocean Large Islands, which include New Zealand and Australia, East Asia, North and West Europe, and Pacific Ocean Small Islands represented between 7.9% and 14.6% of respondents. Africa Atlantic Ocean, Baltic Sea, Caribbean Islands, Northern Mediterranean, South Asia, and Southern Mediterranean made up between 2.0% and 3.2% of the respondents. The Southeast Asia, South America Pacific Ocean, Africa Indian Ocean, South America Atlantic Ocean, and Gulf states had the fewest respondents, each with less than 1%.

At the national scale, we received responses from people in 49 different countries. In forty of the countries we had between one and four respondents. In nine countries we had higher participation. China, Israel, and Japan each had 5 respondents and together they represent 6% of our respondents. In the middle was the United Kingdom, New Zealand, and South Korea with 8, 10, 13 respondents, respectively. Australia, Canada, and the United States had the greatest number of respondents (26, 26, and 94, respectively). Within the broader context illuminated

by the present analysis, we aim to conduct subsequent research activities to better investigate regions such as the Caribbean and Latin America, Africa, and South-east Asia, which were less represented in this research process.

Respondents represented a variety of jurisdictional scales but tended towards a local scale that afforded a unique and tangible perspective on climate adaptation efforts undertaken to directly address SLR threats. 163 respondents (65%) were from local governments (for example, cities, councils, municipalities, towns, and native settlements) with three (1.2%) from infrastructure-specific settings (for example ports, airports, and ferries), and 60 (24.0%) respondents were from sub-national governments (for example districts, provinces, states, and territories). Only 24 (9.6%) of our respondents were from national governments. Sixteen (66%) of the national respondents were from island nations such as Nauru in Oceania and Trinidad and Tobago in the Caribbean while eight (33%) were from continental settings such as Bangladesh in Asia and Liberia in Africa. The high representation from local and sub-national respondents aligns with our objective of understanding the use of climate science by those with direct decision-making authority on infrastructure design and land use.

Respondents represent places that account for over 1 billion people. The places respondents answered for range widely in population. At the local government scale, Monhegan, Maine in the United States is the smallest place that we had a respondent from, with a population of 69. At the other end of the size spectrum, we had a local government response from Tianjin, China with a population of over 13 million. The mean population size for local government respondent is 1.04 million. At the sub-national scale, the largest place represented by a respondent was the State of California in the United States, which has a population of over 39 million. The smallest sub-national respondent was from the Territory of Nunavut—Kugluktuk in Canada, which has a population of 1491. The mean sub-national scale population is 3.7 million. Finally, at the national scale, the respondent from the smallest place was Niue with a population of 1620, and the largest place was Bangladesh with 165 million people. The mean national scale population is 25 million. Gathering data from this wide range of populations allows us to gain insight into different places and their unique approaches to using SLR in planning.

**Survey design**. The questionnaire ran from November 2020 to August 2021, which can be found in Supplementary Information (Appendix 1). The questionnaire was designed for coastal managers across the globe to help us understand publicly available information about places and their management decisions relative to SLR planning. The questionnaire was conducted via an online survey platform, Qualtrics. The survey was written in English by the authors, several of whom are native speakers. The survey was then translated into 8 languages (Arabic, Chinese, French, Hebrew, Japanese, Korean, Portuguese, and Spanish) by professional translators. Native speakers of each language verified the translations.

The questionnaire was divided into four sections and consisted of 22 questions for respondents that are using SLR projections in planning. Section 1 was about the specific place and whether it has formally included future sea levels in its planning processes. Section 2 asked questions about formal local policies that include SLR projections, when the documents were developed, how much regulatory force they have, their specific projections for 2050 and 2100, and whether they consider sea-level projections past 2100. Section 3 asked further questions about the science and physical processes included in the SLR projections used. Section 4 asked questions about the use of the SLR projections, such as whether the projections have affected development plans, what criteria go into the location's decision-making processes, what kind of planning approaches the location uses, and how often the projections are to be updated. For respondents not formally using SLR projections in planning the questionnaire consisted of 2 sections and a total of 5 questions. These respondents had the same first section as those that are using SLR projections. Section 2 asked general questions about coastal planning, hazards, and if they are engaged in a process to start to use SLR projections in the future.

**Analysis**. These data are not well suited to making robust statistical inferences since the snowball sampling method is intentionally non-random and therefore subject to bias. However, the sampling technique does lend itself well to an analysis that is more qualitative in nature. Therefore, the aim here was to present a descriptive overview of the survey data in such a way that it provided insight into our critical research questions. To that end, the analysis primarily focused on where survey respondents are based, the structure of the projections that the respondents are using, and the projection values that they provided. We handled some anomalies in the data by switching zeros to NA in cases where zero didn't make sense relative to other values provided. We also re-labeled 3 structures to reflect the data provided. In conducting the analysis, we spatially assessed the continents (Fig. 1A) and regions (Fig. 1B) with high and low utilization of SLR projections in planning. We considered the breakdown of projection structures (A, B, C, D) conditional on the continent (Fig. 2) and region (Supplementary Fig. 2), to gain insight into the spatial variability of the structures being used. We summarized the projection values within the various projection structures (Fig. 3, Supplementary Fig. 3, Supplementary Tables 5 and 6) to investigate any notable internal consistency and/or external consistency. An example of internal consistency in this context would be seeing similarities in projections when comparing low/high

estimates across structures B, C, and D. An example of external consistency would be seeing that projection ranges within the different structures align with IPCC or SROCC projections (Fig. 3).

**Reporting summary**. Further information on research design is available in the Nature Portfolio Reporting Summary linked to this article.

## Data availability

The data files for producing the maps, tables, and graphs of this manuscript are deposited in the public repository of Utah State University at https://digitalcommons.usu.edu/all_datasets/198/.

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

## Acknowledgements

Many people across the world provided critical information to our questionnaire. We would like to thank them for taking time out of their busy schedules to support this research. D.H. was funded by Utah State University's Office of Research. D.B. was funded by the people of the City and County of San Francisco and the SFPUC to participate in this research. R.J.N. was supported by the PROTECT Project. This project has received funding from the European Union's Horizon 2020 research and innovation program under grant agreement number 869304, PROTECT contribution number 54. NC's work was conducted with the financial support of Science Foundation Ireland and co-funded by Geological Survey Ireland under grant number 20/FFP-P/8610. For T.J., this is a contribution of the Climate Change Geoscience Program of Natural Resources Canada. BPH is supported by the Singapore Ministry of Education Academic Research Fund MOE2019-T3-1-004, the National Research Foundation Singapore, and the Singapore Ministry of Education, under the Research Centres of Excellence initiative. This work is Earth Observatory of Singapore contribution 499. R.G.B. was supported by the NZ SeaRise Program funded by New Zealand Ministry of Business, Innovation & Employment Contract to the Research Trust at Victoria University (Contract ID - RTVU1705). M.C. acknowledges support from the US Climate Variability and Predictability (CLIVAR) program. For M.E., the present work was performed as a part of activities of Research Institute of Sustainable Future Society, Waseda Research Institute for Science and Engineering, Waseda University. K.M. was supported by the Climate Systems Hub of the Australian Government's National Environmental science Program (NESP) and CSIRO. We would like to recognize Muhammad Hadi Ikhsan for his work on the figures.

## Author contributions

D.H.: conceptualization, methodology, resources, data curation, writing—original draft, writing—review & editing, supervision, project administration, funding acquisition. D.B.: conceptualization, methodology, writing—original draft, writing—review & editing, project administration. R.N.: conceptualization, methodology, writing—original draft, writing—review & editing, supervision, project administration, funding acquisition. N.C.: formal analysis, writing—review & editing, visualization. R.B., M.C.: investigation, methodology, writing—review & editing. M.E., B.G.: investigation, writing—review & editing. B.H: investigation, writing—review & editing, supervision. T.J.: investigation, methodology, resources, writing—review & editing, supervision. M.E.P.: investigation, methodology, writing—review & editing, supervision. M.R.: investigation, writing—review & editing. O.B., H.L.: methodology, investigation. S.K.: investigation, data curation, visualization. K.A.A., F.A., M.A., A.B., R.B., J.F, A.G., S.H., M.J.K., K.M., N.M., D.R., L.W., H.Y.: investigation.

## Competing interests

The authors declare no competing interests.

**Additional information**

