## [Peer Review File · Communications Earth & Environment]

23rd Aug 22

Dear Dr Hirschfeld,

Your manuscript titled "A Global Survey of the application of Sea-Level Projections" has now been seen by 3 reviewers, and I include their comments at the end of this message. They find your work of interest, but some important points are raised. We are interested in the possibility of publishing your study in Communications Earth & Environment, but would like to consider your responses to these concerns and assess a revised manuscript before we make a final decision on publication.

We therefore invite you to revise and resubmit your manuscript, along with a point-by-point response that takes into account the points raised. In particular, we ask that you consider and address the points raised by Reviewers #1 and #2 regarding how the context of coastal adaptation to local sea level rise may differ between regions and practitioners. We also ask that you provide more detail on your snowball sampling approach, as requested by Reviewer #3. Please highlight all changes in the manuscript text file.

Please use the following link to submit your revised manuscript, point-by-point response to the referees' comments (which should be in a separate document to any cover letter) and the completed checklist:

[link redacted]

We hope to receive your revised paper within six weeks; please let us know if you aren't able to submit it within this time so that we can discuss how best to proceed. If we don't hear from you, and the revision process takes significantly longer, we may close your file. In this event, we will still be happy to reconsider your paper at a later date, as long as nothing similar has been accepted for publication at Communications Earth & Environment or published elsewhere in the meantime.

We understand that due to the current global situation, the time required for revision may be longer than usual. We would appreciate it if you could keep us informed about an estimated timescale for resubmission, to facilitate our planning. Of course, if you are unable to estimate, we are happy to accommodate necessary extensions nevertheless.

Please do not hesitate to contact me if you have any questions or would like to discuss these revisions further. We look forward to seeing the revised manuscript and thank you for the opportunity to review your work.

Best regards,

Clare

Clare Davis, PhD
Senior Editor
Communications Earth & Environment

www.nature.com/commsenv/
@CommsEarth

EDITORIAL POLICIES AND FORMATTING

Editorial Policy: [Policy requirements](https://www.nature.com/documents/nr-editorial-policy-checklist.zip)

Furthermore, please align your manuscript with our format requirements, which are summarized on the following checklist:

[Communications Earth & Environment formatting checklist](https://www.nature.com/documents/commsj-phys-style-formatting-checklist-article.pdf)

and also in our style and formatting guide [Communications Earth & Environment formatting guide](https://www.nature.com/documents/commsj-phys-style-formatting-guide-accept.pdf) .

***** DATA:** Communications Earth & Environment endorses the principles of the Enabling FAIR data project (<http://www.copdess.org/enabling-fair-data-project/>). We ask authors to make the data that support their conclusions available in permanent, publically accessible data repositories. (Please contact the editor if you are unable to make your data available).

All Communications Earth & Environment manuscripts must include a section titled "Data Availability" at the end of the Methods section or main text (if no Methods). More information on this policy, is available at <http://www.nature.com/authors/policies/data/data-availability-statements-data-citations.pdf>.

If a community resource is unavailable, data can be submitted to generalist repositories such as [figshare](https://figshare.com/) or [Dryad Digital Repository](http://datadryad.org/). Please provide a unique identifier for the data (for example a DOI or a permanent URL) in the data availability statement, if possible. If the repository does not provide identifiers, we encourage authors to supply the search terms that will return the data. For data that have been obtained from publically available sources, please provide a URL and the specific data product name in the data availability statement. Data with a DOI should be further cited in the methods reference section.

REVIEWER COMMENTS:

Reviewer #1 (Remarks to the Author):

Review of 'A Global Survey of the Application of Sea-Level Projections'

In this manuscript, the author(s) discuss the results of a survey about how practitioners use sea-level projections. They conclude that not all practitioners use sea-level rise in their planning and sometimes a limited number of scenarios. The study is interesting, well-written, and could help sea-level scientist shape their narrative. However, I have a few major questions/remarks about the setup.

The first thing I wonder about is for what purpose the practitioners who filled in the questionnaire are using sea-level projections. In other words: 'what do they do with them?'. That is very important to know in this context, because the use case largely dictates how they can/should use the projections. Now, every practitioner is thrown on the same heap, while in practice, they all need a different projection to fit their need. Some examples on how their needs can differ:

- Time scales: in the current manuscript, there's no discussion on the planning horizon used by the practitioners. For example, practitioners who want to know how much sand must be supplied annually to retain a sandy beach don't need to care about long-term scenarios and high-end estimates, since the latter only emerge in about 50 years or so (e.g. Kopp et al. 2017).

- If a practitioner needs to make a very risk-averse decision (for example finding a location for a nuclear power station or hospital) and goes with a single high-end scenario, the rationale is much different from a practitioner who selects a single scenario without regard for the spreads and uncertainties.

A discussion about the applications is needed, because throughout the paper, it is suggested that there are 'good' and 'not-so-good' approaches for selecting projections. For example: "While recognition of the threat of SLR is almost universally recognized, only 71% of respondents currently utilize SLR projections." In the abstract and "We found surprisingly that most coastal managers are using a single SLR projection rather than the recommended approach of considering a range of possible SLR values to account for uncertainty. We also learned that a wide range of future projections are in use and that there is no globally standardized approach to selecting and using SLR projections." on line 66. But without understanding what the projections are used for, such statements are not valid in my opinion. Hence, I think it is necessary to add a thorough discussion for what purpose the interviewed practitioners use the projections and in what time scales they are interested.

What is possibly also interesting to discuss is the extent in which practitioners use regional/local versus global-mean projections. It is important to note that most projection products, such as IPCC AR6/SROCC/AR6 and many local assessments explicitly take into account the regional deviations from GMSL projections due to GIA/contemporary GRD/Ocean dynamics and sometimes coastal subsidence. Are these local estimates used by practitioners?

Finally, what might also be interesting (and part of the provided questionnaire) to discuss: what source is used for the projections: for example, do many practitioners rely on specific papers or do they use IPCC-type assessments?

Line-by-line comments

L12-14: "This research proves insightful for improving sea-level science, and informs important ongoing efforts on the application of the science which are essential to promote effective adaptation." Where is the evidence for this statement? I think it is unnecessary to state this anyway and remove this sentence.

L37: 'Deep uncertainty' is a very specific term about uncertainties that cannot be quantified due to a lack of understanding of the underlying processes. It is something completely different than 'large uncertainty'. I'd suggest to use the latter here.

L72-L73: Here, I think we desperately need information about what kind of problems and projects these coastal managers work on.

L135: This 'discrepancy' cannot be regarded a discrepancy without discussing the specific use cases. For generic nation-wide assessments, a multi-scenario approach that includes some high-end scenarios fully make sense and can indeed be the most appropriate scenario, but for many more specific cases this is not necessarily the case.

L147: "Hence, the survey identifies that more guidance concerning the use of high-end

scenarios, including adaptive decision analysis, would be useful.” How does this statement follow from the survey, and what kind of guidance beyond what’s already available would be useful?

L181: Note here that the projections from the AR5 and SROCC reports did not include low-confidence processes in their RCP8.5 scenarios. In for example AR6, there are specific low-confidence/high impact scenarios for various RCP8.5 / SSP5-8.5 emissions scenarios.

L188: One of the possible causes of this difference is the large deviation of local sea-level projections from their GMSL counterparts. Check for example Yin et al. (2009) of why sea the US east coast (with many survey respondents) will face a much larger MSL rise compared to GMSL. One option to reflect this in Figure 3 is to incorporate the total local projection spread in AR5/SROCC instead of the GMSL projection, for example by taking the 5th/95th percentile of the gridded projections instead of the global-mean projections.

L217: Akin to above: the ‘singular’ scenario could have taken this uncertainty into account, for example by accounting for an allowance for the uncertainty. For many adaptation measures, you need a single number. For example, if an engineer wants to build a new seawall, he/she will need one single design height, and using a whole set of possible future sea levels is unusable.

References

Kopp, R. E., DeConto, R. M., Bader, D. A., Hay, C. C., Horton, R. M., Kulp, S., Oppenheimer, M., Pollard, D., & Strauss, B. H. (2017). Evolving Understanding of Antarctic Ice-Sheet Physics and Ambiguity in Probabilistic Sea-Level Projections. *Earth’s Future*, 5(12), 1217–1233. <https://doi.org/10.1002/2017EF000663>

Yin, J., Schlesinger, M. E., & Stouffer, R. J. (2009). Model projections of rapid sea-level rise on the northeast coast of the United States. *Nature Geoscience*, 2(4), 262–266. <https://doi.org/10.1038/ngeo462>

Reviewer #2 (Remarks to the Author):

General

This is an excellent, original and timely study that makes a valuable contribution to decision making for climate change adaptation. There is indeed an urgent need to understand the extent to which coastal decision makers are using SLR projections and importantly the types of projections they are using. That the authors have investigated this at a global scale and present data comparatively and by region is also a considerable contribution.

The study design is robust: targeted sampling and qualitative analysis are appropriate for this context and the breadth of responses is impressive. It is noteworthy that the authors sought

respondents at sub-national and local scales as this is where decision-making on coastal management occurs (in the main) and is a fact missed by many studies in the design of their methods.

The findings of the paper are clearly communicated and well-pitched to the audience of the journal. The figures are useful and well-presented, and the description of methods and supplementary material was also useful. I would have liked to see more detail but I note that this is an overview paper and the restrictions of the article type - I hope to see more published from this study.

Overall, I thought this was an excellent paper and it is rare that I have so little to add in a review. Below are two more specific suggestions that I am not wedded to and should be considered at the editor and authors discretion. I look forward to seeing this published!

Specific

An underlying point to this study is the relative urgency and differential consequences for different regions in making decisions about the nature and types of adaptation in the context of SLR projections. The authors have picked out some countries to discuss in more detail explanations for the structure of the projections they use – the example of Japan added context - It would be good to make the point about relative vulnerability briefly. I.e. for some countries the consequences of using projections that are not suitable (single or low and high end) are higher. An example of this would be good. Although I also understand that this discussion requires nuance and may be better articulated in another paper.

In line with the above comment – the points made in lines 142-147 are crucial to the implications for policy makers and it would be good to see these reiterated in the conclusion. The second point about the dangers of using H++ projections is especially important in the context of many small island states and atolls where the use of those projections would essentially limit adaptation planning to one option (migration) if used as a single projection in the 2050 time horizon. This matters and is a point that is not well understood.

Reviewer #3 (Remarks to the Author):

Thank you so much for the opportunity to review this work. I have no understanding of how to assess/think about sea level projections. However, the editors have asked me to review the part of this work around snowball sampling which I have some insight on, I hope. Upon reviewing this work, I feel that, indeed, more specification is needed in this part. I have authored a work on enhancing sample diversity in the environmental sciences, published with Katrina Childs in PLOS ONE, which may be of interest to this work. Link to this work: <https://journals.plos.org/plosone/article/authors?id=10.1371/journal.pone.0201710>. Our paper is all about enhancing sample diversity which seems to be an issue with this paper. The authors themselves note that the 253 coastal managers responded to the questionnaire have limited geographic diversity, for instance. I encourage the authors to further elaborate what efforts they have undertaken in order to enhance the diversity of their sample (and to also

bring in the term 'sample diversity'). For instance, have they attempted to improve seed diversity? How persistent have they been in securing interviews? Have they leveraged all of their personal contacts? Have they considered not only undertaking email interviews, but also face-to-face ones which tend to also boost sample diversity? I find that emphasizing that (hopefully) various strategies have been undertaken to improve sample diversity would help to enhance the robustness of this work. If no such strategies were undertaken, this could at least be noted as an avenue for future research. As of now, it is not clear what has been done to ensure sample diversity. Generally, it is good to publish research even if sample diversity is limited, as long as all avenues of boosting it have been taken.

Response to Reviewers at Communications Earth & Environment COMMSENV-22-0539-T

Title: A Global Survey of the Application of Sea-Level Projections

Dear Reviewers,

Thank you very much for your time and your thoughtful comments. Your insights helped to improve our paper. We made edits to our manuscript and want to be sure your concerns are thoroughly addressed. We agree that several factors – including practitioner use, local vulnerability, information sources, etc. – could drive the differences we observed in our responses. We recognize the issues with our sample diversity and provided details on our efforts to enhance this diversity. Below we address this in greater detail.

We provided the track changes version (from the first draft) that highlights where we made edits in response to your suggestions. We also provided a clean version with track changes accepted as this may be easier to read. To make our responses easy to follow, the table below lists each comment, provides our response, and points to the place in the text where you will find the edits made.

No.	Comments by reviewers (paraphrased)	Response	Where changed
Reviewer #1			
1.	I think it is necessary to add a thorough discussion for what purpose the interviewed practitioners use the projections and in what time scales they are interested.	We agree that this is very important and highly relevant to improving this research. We expanded the discussion to call attention more clearly to the need to understand use cases. We acknowledge that planners and engineers could use different numbers. We also recognize that different numbers are useful in different decisions contexts. For example, the SLR projection needed for building a hospital are different than the needs for prioritizing the conservation efforts around a mangrove that could be a nature-based solution for SLR.	Content added to the discussion section
2.	What is possibly also interesting to discuss is the extent in which practitioners use regional/local versus global-mean projections.	Our research does not allow us to easily answer this question. We added a discussion of this topic and mention the need to better address this in future surveys.	Content added to the discussion section

3.	What source is used for the projections: for example, do many practitioners rely on specific papers or do they use IPCC-type assessments?	We added a supplementary figure #4, which shows that practitioners relied on three ways to get their projections: 1) Selected from projections, 2) Co-produced, and 3) Generated as guidance by an authority. This indicates that climate services or translation services are heavily involved in the process. We added some discussion around this topic.	Content added to the discussion section Figure added to supplementary text.
----	---	---	--

Reviewer 1 – smaller line by line comments			
4.	L12-14: “This research proves insightful for improving sea-level science, and informs important ongoing efforts on the application of the science which are essential to promote effective adaptation.” Where is the evidence for this statement? I think it is unnecessary to state this anyway and remove this sentence.	We’ve changed the sentence to more accurately reflect the insights from our research.	We revised this concluding sentence in the abstract.
5.	L37: ‘Deep uncertainty’ is a very specific term about uncertainties that cannot be quantified due to a lack of understanding of the underlying processes. It is something completely different than ‘large uncertainty’. I’d suggest to use the latter here.	Agreed	Changed in line 37
6.	L72-L73: Here, I think we desperately need information about what kind of problems and projects these coastal managers work on	We added some additional information on the scale of government and the general nature of our respondents. However, we acknowledge that as a non-human subjects research project we are not able to dive in too deeply to the specifics of our respondents. Future work will more deeply consider use cases and the specific decision-making contexts.	Changed in first paragraph of results.
7.	L135: This ‘discrepancy’ cannot be regarded a discrepancy without discussing the specific use cases. For generic nation-wide assessments, a multi-scenario approach that includes some high-end scenarios fully make sense and can indeed be	We agree that in some use cases a singular number could be appropriate, however this survey was not sent to engineers or designers building a specific thing. These are numbers that are used for planning in 2100 (and 2050 as presented in Supplementary Information). While a single number could be appropriate for an engineering project, it is not appropriate for	We made change to the discussion to acknowledge the specifics of use cases.

	the most appropriate scenario, but for many more specific cases this is not necessarily the case.	long-term planning. Even for long-term planning for engineering projects they require adaptive management in a non-stationary world. We would note also that uncertainties don't only exist for high end SLR, but for the "likely" long term range (e.g. 17-83% range in IPCC reports), for mid-century when projections are relatively unimpacted by emissions uncertainty but are by sensitivity uncertainty. In this environment, prudent planning requires consideration of ranges for nearly all time frames rather than a single number.	
8.	L147: "Hence, the survey identifies that more guidance concerning the use of high-end scenarios, including adaptive decision analysis, would be useful." How does this statement follow from the survey, and what kind of guidance beyond what's already available would be useful?	True that it does not come directly from the survey. It comes from the previous sentence. Language has been changed to reflect that. As for the type of guidance, we think the guidance in van de Wal et al could be beneficial.	
9.	L181: Note here that the projections from the AR5 and SROCC reports did not include low-confidence processes in their RCP8.5 scenarios. In for example AR6, there are specific low-confidence/high impact scenarios for various RCP8.5 / SSP5-8.5 emissions scenarios.	Noted.	We did not make a change
10.	L188: One of the possible causes of this difference is the large deviation of local sea-level projections from their GMSL counterparts. Check for example Yin et al. (2009) of why sea [level rise on] the US east coast (with many survey respondents) will face a much larger MSL rise compared to GMSL. One option to reflect this in Figure 3 is to incorporate the total local projection spread in AR5/SROCC instead of the GMSL projection, for example by taking the 5th/95th percentile of the gridded projections instead of the global-mean projections.	This is an excellent point but is beyond the scope of this first paper. To do this we would need to provide gridded RSLR figures for each of our 247 respondents and then compare across the sample. Apart from methodological issues (what gridded data was available c. 2019, and were practitioners even aware of this gridded data, which is much more obscure than IPCC reports), this would require a different level of analysis that would best be undertaken in future work.	

11.	L217: Akin to above: the 'singular' scenario could have taken this uncertainty into account, for example by accounting for an allowance for the uncertainty. For many adaptation measures, you need a single number. For example, if an engineer wants to build a new seawall, he/she will need one single design height, and using a whole set of possible future sea levels is unusable.	Noted. We changed the text to take this point into account.	In concluding paragraph
-----	--	---	-------------------------

REVIEWER #2			
1.	Make the point about relative vulnerability in different places.	This is a good point and we acknowledge this in the discussion	
2.	The points made in lines 142-147 are crucial to the implications for policy makers and it would be good to see these reiterated in the conclusion	Agreed	Added content to conclusion

REVIEWER #3 – Methods

1.	Further elaborate what efforts they have undertaken to enhance the diversity of their sample. Add information on what approaches were taken, such as: persistence, leveraging contacts, using interviews and not just emails, etc.	We used all five of the methodological recommendations articulated in Kirchherr & Charles. Specific steps that we took include: 1. We relied heavily on personal contacts. For example, we had a regional resilience coordinator in the Caribbean and the Pacific Islands. In total we had 23 collaborators to help develop our list of contacts.2. We had a diverse seeding process reaching out to multiple people in a single region.3. We worked hard to develop trust with our respondents and referrers. We made personal phone calls to specific individuals to explain our research and help them decide to engage with our work.4. We were very persistent in securing responses. We emailed repeatedly. We also emailed several colleagues for a single location to enhance response possibility for those places.5. We had two sampling waves and did focused communications with people in hard to reach regions. In addition to the methodological recommendations, we allowed a range of ways to respond. For example, in China we allowed respondents to complete the questionnaire and then our colleague entered the data into the online platform since firewall protections hindered website access. However, because it was a survey, we ultimately did not allow for face-to-face interviews. We wanted responses to be standardized in our survey platform. We acknowledge that future research should be done to get an even more diverse sample.	We primarily edited the methods section of the manuscript (after the first list of references) to emphasize the work we did do to enhance sample diversity. We also added content into the discussion about the low geographic diversity and the resources needed to support these methodological efforts to increase the diversity of the sample.
----	--	---	--

23rd Nov 22

Dear Dr Hirschfeld,

Your manuscript titled "A Global Survey of the application of Sea-Level Projections" has now been seen by our reviewers, whose comments appear below. In light of their advice I am delighted to say that we are happy, in principle, to publish a suitably revised version in Communications Earth & Environment under the open access CC BY license (Creative Commons Attribution v4.0 International License).

We therefore invite you to revise your paper one last time to address the remaining concerns of our reviewers. In particular, we ask that you report the results of the survey section on "Application of Sea Level Guidance" and discuss these findings in relation to your claims, or appropriately caveat and tone down these claims. At the same time we ask that you edit your manuscript to comply with our format requirements and to maximise the accessibility and therefore the impact of your work.

Please note that it may still be possible for your paper to be published before the end of 2022, but in order to do this we will need you to address these points as quickly as possible so that we can move forward with your paper.

EDITORIAL REQUESTS:

SUBMISSION INFORMATION:

OPEN ACCESS:

Communications Earth & Environment is a fully open access journal. Articles are made freely accessible on publication under a [CC BY license](http://creativecommons.org/licenses/by/4.0) (Creative Commons Attribution 4.0 International License). This license allows maximum dissemination and re-use of open access materials and is preferred by many research funding bodies.

For further information about article processing charges, open access funding, and advice and support from Nature Research, please visit <https://www.nature.com/commsenv/article-processing-charges>

At acceptance, you will be provided with instructions for completing this CC BY license on behalf of all authors. This grants us the necessary permissions to publish your paper. Additionally, you will be asked to declare that all required third party permissions have been obtained, and to provide billing information in order to pay the article-processing charge (APC).

[link redacted]

Best regards,

Clare

Clare Davis, PhD
Senior Editor
Communications Earth & Environment

www.nature.com/commsenv/
@CommsEarth

REVIEWERS' COMMENTS:

Reviewer #1 (Remarks to the Author):

Review of 'A Global Survey of the Application of Sea-Level Projections'

This is the second round of review of this manuscript, and most points have been adequately addressed. However, I am still disappointed in the discussion of the use of projections. Since that use case is in my opinion the prime factor to decide what kind of projection to use. There is a section in the questionnaire about this ('Application of Sea Level Guidance", as well as questions 14 and 15), and I wonder why the responses to these questions are not discussed in the current manuscript.

Given this lack of information on the use, I still find the contrast (now in Line 139ff) between what is used by practitioners versus what is recommended by scientist not convincing. Many practitioners, from engineers who design a dike or lock to councils who draw zoning maps, need to make binary decisions (how high do I build my dike, can I build homes in this plain, is it safe to build a nuclear plant here, etc. etc.) based on uncertain futures, and thus 'need' a single number. This problem is also visible in for example Figure 3, where the median singular scenario assumes a much higher sea level than most middle-of-the-road projections (let's say SROCC RCP4.5).

Therefore, I advise to explicitly include the outcomes of the questions on how the projections have

been used in the manuscript, and link these outcomes to the projection structures that have been used.

Response to Reviewers at Communications Earth & Environment COMMSENV-22-0539-T

Dear Reviewer,

Thank you very much for your time and your thoughtful comments. Your insights helped to improve our paper. We made edits to our manuscript and supplementary materials to address your concerns. Below in the table we list your comments, provide our response, and points to the place in the text where you will find the edits made.

No.	Comments by reviewers (paraphrased)	Response	Where changed
1.	The use case for projections are the prime factor to decide what kind of projection to use. There is a section in the questionnaire about this and I wonder why the responses to these questions are not discussed in the current manuscript	Use cases are important and we would like to relate the use cases to our findings. However, survey respondents identified most use cases and did not clearly link a singular use case with specific projections. We are not able to make a clear relationship in this paper. We did add the use case information that we have and we aim to further research this relationship in future projects.	Survey responses are added to the supplemental materials. We call attention to the use case context in lines 149 & 233.
2.	I find the contrast between what is used by practitioners versus what is recommended by scientist not convincing. Many practitioners, from engineers who design a dike or lock to councils who draw zoning maps, need to make binary decisions (how high do I build my dike, can I build homes in this plain, is it safe to build a nuclear plant here, etc. etc.) based on uncertain futures, and thus 'need' a single number	We agree that a singular number is needed in many contexts as you have accurately described. Our concern is that these numbers should arise out of careful consideration of a range of projections. This range of numbers should be in the planning or guidance materials that are adopted before an investment is made in a new engineered structure or a law is adopted.	We edited the text starting in line 143
3.	I advise to explicitly include the outcomes of the questions on how the projections have been used in the manuscript, and link these outcomes to the projection structures that have been used.	We appreciate your recommendation and aim to achieve these insights in future research. The current project does not allow us to make this connection. We did include the results in our supplementary material.	Survey responses are added to the supplemental materials.